# Rosetum3D: A Large-Scale 3D Vision Dataset from Preharvest Roses

## Abstract

The global rose cultivation industry has experienced continued expansion driven by rising demand for cut flowers and botanical extracts. However, automated pre-harvest quality grading faces significant challenges due to occlusion-heavy canopy environments and morphological variations across growth stages. While 3D computer vision offers solutions for localization tasks, progress has been hindered by the lack of large-scale datasets with phenotypic keypoint annotations that capture plant architecture. To bridge this gap, we introduce **Rosetum3D**, constructed via occlusion-robust multi-view RGB-D capture protocols in commercial greenhouses. We provide fine-grained 2D localization using bounding boxes and botanically defined keypoints with 3D structures recovered through depth backprojection. Models trained on Rosetum3D have achieved 2D/3D rose localization, a crucial step for automated pre-harvest quality grading and growth monitoring. Beyond localization, Rosetum3D serves as a benchmark for agricultural vision tasks, including 2D rose object detection, local feature matching, and depth estimation. By enabling data-driven precision agriculture, Rosetum3D paves the way for robotic harvesting systems and AI-driven yield prediction in protected cultivation.

## 1 Introduction

The global rose industry has experienced significant growth in recent years, driven by increasing demand for both ornamental and commercial applications, including essential oils and cosmetics. Efficient monitoring of pre-harvest roses is crucial for optimizing yield and ensuring quality, particularly for growth monitoring and automated grading. Computer vision technologies, capable of precise 2D and 3D localization of roses, offer a promising solution for this application. However, the development of such systems requires large-scale datasets tailored to the challenges of agricultural environments, where dense foliage and complex plant structures complicate data acquisition and annotation.

To address this gap, we introduce **Rosetum3D** in this paper, a dataset collected using structured-light RGB-D cameras in operational rose greenhouses, as shown in Figure 1. Structured-light sensors were chosen over other 3D sensors, like LiDAR, for their cost-effectiveness and robustness under typical greenhouse lighting conditions. Multi-view RGB-D sequences were captured across diverse rose varieties and growth stages, ensuring coverage of variations in plant geometry and occlusion patterns.

Despite significant advances in 3D vision, existing public datasets (e.g., KITTI (Geiger et al., 2012), nuScenes (Caesar et al., 2020), ScanNet (Dai et al., 2017), SUN RGB-D (Song et al., 2015)) focus on autonomous driving or indoor scenes, where LiDAR or photogrammetric 3D reconstructions enable direct annotations on point clouds or meshes. However, these approaches falter in rose cultivation settings due to severe inter-plant occlusions, fragile petals prone to deformation, and dynamic lighting conditions. As a result, annotations for Rosetum3D are generated through a hybrid 2D-to-3D pipeline. First, 2D bounding boxes and semantic keypoints (e.g., corolla center, sepal-pedicel junction, stem-pedicel transition, soil-emergence point, etc.) are manually labeled on RGB images. These annotations are then re-projected into 3D space using depth maps from synchronized RGB-D frames. This approach bypasses the need for error-prone 3D reconstruction while preserving geometric consistency across viewpoints.

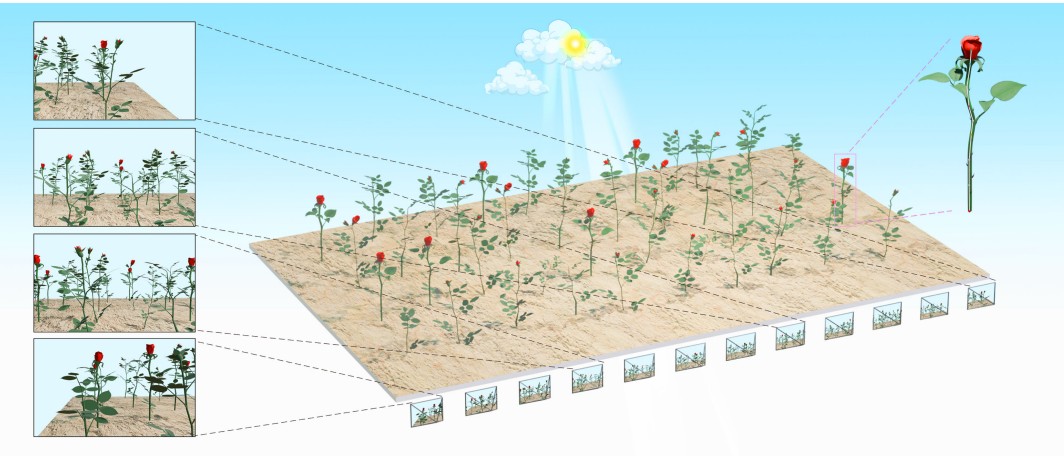

Figure 1: Illustration of Rosetum3D. The users walk parallel to the plant rows at a constant speed and take RGB-D sequences of the preharvest roses. After collection, the annotators mark the bounding boxes and keypoints on the 2D images, and the annotations are backprojected to the 3D space to compute the 3D coordinates. The 3D keypoints are shown in the top-right corner.

Beyond supporting core tasks such as 2D/3D localization of unharvested roses, Rosetum3D serves as a versatile benchmark for broader computer vision challenges. Its multi-view RGB-D sequences enable the evaluation of local feature matching algorithms, as well as monocular and multi-view depth estimation under heavily occluded situations. By bridging the lack of agriculturally relevant 3D data, Rosetum3D aims to advance both task-specific solutions and fundamental vision research in complex, unstructured settings.

In summary, the contributions of this paper include:

- We propose Rosetum3D, the first large-scale 3D visual dataset for rose cultivation scenarios, comprising synchronized multi-view RGB-D sequences and meticulously annotated 2D/3D labels (bounding boxes, keypoints) for unharvested roses.
- Through extensive experiments, we demonstrate that Rosetum3D establishes a novel evaluation benchmark for diverse vision tasks, including 2D object detection, 2D/3D keypoint localization, local feature matching, and monocular/multi-view depth estimation in occlusion-heavy agricultural environments.
- Rosetum3D provides a foundational platform for developing and testing vision algorithms tailored to agricultural automation, particularly for robotic harvesting, yield prediction, and plant phenotyping in rose cultivation systems.

## 2 RELATED WORK

### 2.1 3D VISION DATASET

The acquisition of accurate depth supervision is fundamental for 3D computer vision, as monocular images inherently lack depth information. Consequently, 3D datasets typically employ specialized sensors, such as LiDARs and RGB-D cameras, to obtain ground-truth geometry. This sensor dependency has led to the development of domain-specific dataset paradigms.

**Indoor Scenes** primarily leverage RGB-D sensors for dense depth estimation. The NYU Depth Dataset V2 (Silberman et al., 2012) provides aligned RGB-D frames that support 3D segmentation and object recognition. Scaling this effort, the SUN RGB-D Dataset (Song et al., 2015) provides over 10,000 RGB-D images annotated with 146,617 2D polygons and 58,657 3D bounding boxes, enabling the classification, detection, and semantic segmentation of indoor scenes. ScanNet (Dai et al., 2017) significantly advances indoor reconstruction, featuring over 1,500 scenes and 2.5 million views, which facilitates camera pose estimation, surface reconstruction, and semantic instance segmentation. Its enhanced successor ScanNet++ (Yeshwanth et al., 2023) further improves data

quality and diversity for novel view synthesis and holistic 3D understanding. Additionally, complementary efforts include motion-capture-driven human datasets, such as Human3.6M (Ionescu et al., 2014), which provide precise 3D keypoints for articulated poses.

**Outdoor scenes**, such as Autonomous Driving, dominantly utilize LiDARs for long-range depth capture. The seminal KITTI dataset (Geiger et al., 2012), featuring multi-sensor data (stereo cameras, LiDAR, GPS) from vehicle-mounted platforms, has become the benchmark for stereo matching, optical flow, odometry, and 2D/3D object detection/tracking. Despite KITTI's foundational role, newer large-scale datasets have emerged, for example, nuScenes (Caesar et al., 2020; Fong et al., 2022) offers 1,000 driving scenes across Boston and Singapore, featuring 23-class 3D bounding boxes; PandaSet (Xiao et al., 2021) provides 48,000+ camera images and 16,000+ LiDAR sweeps with 28-class detection and 37-class segmentation labels.

**Plant scenes**, Plant3D (Conn et al., 2017a;b; 2019) contains a total of 714 3D laser scans of Tomato, Tobacco, Sorghum, and Arabidopsis plants obtained within 20–30 days of development. The 3D models do not contain color information. ROSE-X (Dutagaci et al., 2020) dataset contains 3D models of 11 rose bush plants acquired through X-ray computer tomography. The voxels in the volumetric models are labeled into three semantic categories: "Leaf", "Stem", and "Flower". Pheno4D (Schunck et al., 2021) is a dataset of 3D point clouds of 7 maize plants and 7 tomato plants. The plants are scanned with a laser scanner at different growth stages, resulting in 244 point clouds. 126 of them are manually annotated with semantic and instance labels. The Soybean-MVS dataset (Sun et al., 2023) is fundamentally different from the rest in terms of data acquisition modality. The plants are captured with an RGB camera in a controlled setup, and their corresponding point clouds are created through multi-view stereo. A total of 102 point cloud models of five different soybean varieties are reconstructed at 13 stages of the whole growth period. Franchetti et al. (2019) collects 2592 data points related to the plant phenotype and 1728 images of the plants, which are used to validate a vision-based plant phenotyping analysis method in indoor vertical farming under artificial lighting. However, these datasets are limited in scale and rely primarily on 3D sensors for data acquisition, which often results in a lack of detailed texture information. In this paper, we employ RGB cameras to capture a large number of high-resolution 2D images rich in fine-grained details. These images, when supplemented by depth sensors, enable the extraction of comprehensive 3D feature representations.

## 2.2 3D VISION IN AGRICULTURE

Different from general indoor and outdoor scenes, **Agricultural applications** face distinct challenges, including small object scales, severe occlusions, and dynamic lighting, which drive diverse sensor strategies. For example, Xiao et al. (2023) utilizes drone-based cross-orbit imaging to reconstruct organ-scale 3D models of crops such as sugar beet and wheat. The Crops3D dataset (Zhu et al., 2024) combines terrestrial laser scanning with structured light, fused via Structure-from-Motion (SfM) and Multi-View Stereo (MVS), to reconstruct 8 crop types in field conditions. PlantGaussian (Shen et al., 2025) adopts 3D Gaussian Splatting for spatio-temporal reconstruction of wheat and tobacco under indoor/outdoor settings.

Despite these advances, large-scale 3D datasets remain scarce in agriculture compared to the general indoor and outdoor domains. Existing efforts often focus on small-scale reconstruction or struggle with environments that are occlusion-heavy. Critically, there are no large-scale, dedicated datasets for floriculture applications, such as rose cultivation, where intricate plant structures and dense foliage require specialized 3D representation. Our work bridges this gap by introducing a large-scale, occlusion-robust dataset tailored to rose phenotyping and robotic harvesting.

## 3 ROSETUM3D DATASET

In this section, we provide a comprehensive overview of Rosetum3D, with a focus on dataset construction, including data collection progress, data annotation, and dataset statistics.

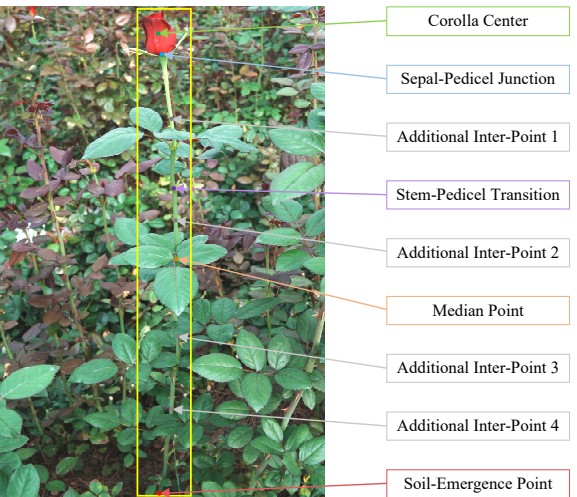

Figure 2: Illustration of 2D Annotation. The main 5 points (*Corolla Center*, *Sepal-Pedicel Junction*, *Stem-Pedicel Transition*, *Median Point*, and *Soil-Emergence Point*) are as described in the main body. Besides those 5 points, we design 4 additional inter-points as follows: *Additional Inter-Point 1* is the midpoint between *Sepal-Pedicel Junction* and *Stem-Pedicel Transition*; *Additional Inter-Point 2* is the midpoint between *Stem-Pedicel Transition* and *Median Point*; *Additional Inter-Point 3* and *Additional Inter-Point 4* are the upper and lower third points between *Median Point* and *Soil-Emergence Point*.

### 3.1 DATA COLLECTION

#### 3.1.1 SENSORS

We use the Orbbec Gemini 335L (Orbbec, 2025a) and 336L (Orbbec, 2025b) to collect the RGB-D sequences, which are stereo vision 3D cameras that combine active and passive stereo vision technologies for seamless operation in both indoor and outdoor conditions. We attach the sensor to a handheld device such as a Windows laptop, enabling the sensor to provide synchronized depth and color capture at 30 Hz. Both the depth and color frames are captured at a resolution of $1280 \times 800$ pixels. We enable auto white balance and auto exposure by default.

#### 3.1.2 CALIBRATION

Here, we need the intrinsic parameters of the camera to reproduce the 3D location of the roses. Similar to ScanNet (Dai et al., 2017), before data collection, the user needs to print out a checkerboard pattern, place it on a large, flat surface, and capture an RGB-D sequence viewing the surface from close to far away. Then we run a calibration procedure based on some existing methods (Teichman et al., 2013; Di Cicco et al., 2015) to obtain intrinsic parameters for both depth and color sensors, and an extrinsic transformation of depth to color.

#### 3.1.3 DATA COLLECTION

The RGB-D data collection was conducted in operational rose greenhouses following a systematic protocol to ensure consistency and coverage of diverse growth stages. As shown in Figure 1, the users walk parallel to the plant rows at a constant speed of $0.3 – 0.5$ m/s, maintaining a fixed distance of $0.8 – 1.2$ meters between the camera and the roses while collecting the data. The camera's tilt angle was calibrated to $15°$ downward from the horizontal plane to optimize coverage of both upper floral regions and lower stem structures. Some examples of the images in the dataset are shown in the Appendix A.

## 3.2 DATA ANNOTATION

The annotation pipeline employs a structured approach inspired by human pose estimation keypoint frameworks, adapted to characterize the morphology of unharvested roses. Each rose instance undergoes the 2D annotation and the 3D reconstruction.

### 3.2.1 2D ANNOTATION

Annotators first mark a bounding box enclosing the entire rose specimen in the RGB frames. After that, they should manually label the following 5 biologically significant keypoints:

- **Corolla Center**: Geometric centroid of the corolla.
- **Sepal-Pedicel Junction**: The basal whorl of sepals converges to the proximal terminus of the pedicel.
- **Stem-Pedicel Transition**: Location where primary stem diameter decreases.
- **Median Point**: The spatial midpoint along the vertical axis of the entire above-ground structure.
- **Soil-Emergence Point**: The precise position where a plant's stem initially penetrates the soil surface.

Besides these 5 points, additional points are annotated between distant keypoints to ensure reliable spatial interpolation, as detailed in Figure 2. In the annotation procedure, if a point is invisible, we will attach an "invisible" label to it, and it will not be taken into account when computing the accuracy. Furthermore, we exclude rose objects smaller than $32 \times 32$ pixels from annotation due to the impracticality of keypoint measurement at such scales, as established in our data quality protocol.

### 3.2.2 3D RECONSTRUCTION

Using synchronized depth maps from RGB-D frames, all 2D keypoints are backprojected into 3D camera coordinates via perspective geometry:

$$\mathbf{P}_{3D} = \mathbf{K}^{-1} \cdot [u, v, 1]^T \cdot D(u, v) \tag{1}$$

where $\mathbf{K}$ denotes the camera intrinsic matrix, $(u, v)$ the 2D keypoint position, and $D(u, v)$ the corresponding depth value.

### 3.2.3 ANNOTATION VALIDATION

To validate the reconstruction accuracy, we capture a subset of scenes containing calibrated rulers placed adjacent to rose specimens. Following the same annotation protocol, we mark two reference points at known distance intervals (e.g., 10 cm) on each ruler within the 2D images. These annotations undergo identical depth backprojection processing to compute their 3D Euclidean distance. A comparative analysis revealed a mean absolute error of 0.14 cm between the reconstructed distances and physical ground truth measurements across 137 test cases, confirming the reliability of our annotation framework in measurement accuracy. The settings and results of this experiment are detailed in the Appendix A.1.

## 3.3 DATA STATISTICS

In total, we collect 20 scenes featuring various types of roses. For these images, we have 46,848 rose objects with the 2D bounding boxes and 2D & 3D keypoint annotations. We select 17 scenes as the training set and 3 scenes as the testing set, details are shown in Table 1. Besides, we conduct a comparative analysis between Rosetum3D and existing public 3D vision datasets in the Appendix A.2.

## 4 TASKS AND BENCHMARKS

Rosetum3D serves as a multi-task benchmark for agricultural perception challenges, including: (1) 2D object detection, (2) 2D rose keypoint localization, (3) monocular/multi-view depth estimation,

| set | Scenes | Images | Objects | Keypoints | Visible-k | Occluded-k | Invisible-k |
|---|---|---|---|---|---|---|---|
| Train | 17 | 17924 | 40759 | 366831 | 281779 | 1758 | 83294 |
| Test | 3 | 3190 | 6089 | 54801 | 42728 | 610 | 11463 |
| Total | 20 | 21114 | 46848 | 421632 | 324507 | 2368 | 94757 |

Table 1: Data Staistics of the Rosetum3D dataset.

| Algorithm | Backbone | AP | $AP_{50}$ | $AP_{75}$ | $AP_M$ | $AP_L$ |
|---|---|---|---|---|---|---|
| Faster R-CNN (Ren et al., 2017) | ResNet50 | 31.0 | 66.8 | 24.7 | 13.1 | 32.1 |
| DETR (Carion et al., 2020) | ResNet50 | 33.0 | 66.5 | 28.8 | 14.2 | 34.3 |
| YOLOv5 (LLC, 2020) | CSPDarkNet53 | **35.0** | **67.1** | **32.4** | **19.1** | **35.9** |
| RT-DETR (Zhao et al., 2023b) | ResNet50 | 33.6 | 64.8 | 30.7 | 13.8 | 34.7 |

Table 2: Experimental 2D object detection results on the Rosetum3D dataset.

and (4) 3D rose keypoint localization. Additionally, we design experiments for local feature matching, and the experimental settings and results are presented in the Appendix A.3. All of these tasks can help with the growth monitoring and automated grading of preharvest roses. We establish baselines using domain-relevant models, including both general state-of-the-art architectures and agriculture-specialized networks, to evaluate under zero-shot and fine-tuned protocols. For 2D/3D rose keypoint localization tasks, we introduce novel stem-aware geometric metrics that utilize adaptive line segmentation to overcome the limitations of traditional point-based metrics.

### 4.1 2D ROSE OBJECT DETECTION

We evaluate several representative object detectors on the Rosetum3D dataset to establish a comprehensive benchmark for object detection. These include: (1) CNN-based detectors, such as Faster R-CNN (Ren et al., 2017), and YOLO-style detectors like YOLOv5 (LLC, 2020); and (2) Transformer-based detectors, such as DETR (Carion et al., 2020). The results are shown in Table 2. We have different evaluation indices for objects of various sizes, which are the same as those in the MSCOCO Dataset (Lin et al., 2014). However, we do not label the small rose objects because they are too far from the camera, making their keypoints difficult to measure. As a result, we show $AP_M$ and $AP_L$ in Table 2. $AP_M$ specifically evaluates the detection performance on medium-sized objects (with an area between $32 \times 32$ and $96 \times 96$ pixels), and $AP_L$ assesses the performance on large objects (with an area greater than $96 \times 96$ pixels).

### 4.2 2D ROSE KEYPOINT LOCALIZATION

To establish a benchmark dataset for 2D rose keypoint localization, we evaluate our proposed dataset using the MMPose framework (Contributors, 2020). We select several high-performing methods originally developed for human pose estimation and transfer them into the rose target, including: (1) Top-down approaches such as RTMPose (Jiang et al., 2023), ViTPose (Xu et al., 2022) All detectors used in the top-down pipeline are based on YOLOv5 (LLC, 2020). (2) Bottom-up approaches such as HRNet (Sun et al., 2019). The experimental results are presented in Table 3. Some qualitative comparison results are shown in the Appendix A.4.

Here, the performance of top-down approaches is inferior to that of bottom-up approaches (e.g., 21.9 AP (VitPose) vs. 24.9 AP (DEKR$^{\dagger}$ + HRNet)). The reason is that the detector limits the performance of top-down approaches. If we replace the detector results with the ground truth, the performance of top-down approaches is significantly better than that of bottom-up approaches (42.1 AP vs. 24.9 AP).

### 4.2.1 LKS-BASED EVALUATION INDEX

Due to the differences between the rose targets and the human targets, the original OKS-based evaluation metric does not adequately capture model performance on the Rosetum3D dataset. Therefore,

| | Method | Backbone | Detector | AP | $AP_{50}$ | $AP_{75}$ | AR | $AR_{50}$ |
|---|---|---|---|---|---|---|---|---|
| Top-down | RTMPose (Jiang et al., 2023) | CSPNeXt-L | YOLOv5 | 16.6 | 45.2 | 8.6 | 27.1 | 54.9 |
| | | | Ground Truth | 37.2 | 82.2 | 28.9 | 48.0 | 86.5 |
| Top-down | ViTPose (Xu et al., 2022) | ViT-L | YOLOv5 | 21.9 | 48.5 | 17.2 | 32.0 | 56.9 |
| | | | Ground Truth | **42.1** | **85.8** | **37.2** | 53.6 | **90.7** |
| Bottom-up | YOLOXPose (Maji et al., 2022) | CSPDarknet | - | *25.0* | *60.3* | *19.3* | **53.7** | *88.7* |
| Bottom-up | DEKR$^\dagger$ + HRNet (Sun et al., 2019) | HRNet | - | 24.9 | 57.8 | 18.6 | 43.9 | 81.2 |

Table 3: Experimental 2D rose keypoint localization results on the Rosetum3D dataset. All top-down methods utilize a YOLOv5 detector, which achieves the best results in Table 2. DEKR$^\dagger$ is an internal design inspired by recent decoupled keypoint regression strategies (Geng et al., 2021). *Italic* indicates the best results without extra information. **Bold** indicates the best results by introducing extra information (the ground-truth detection boxes).

| Method | Detector | AP@1° | AR@1° | AP@3° | AR@3° | AP@10° | AR@10° |
|---|---|---|---|---|---|---|---|
| RTMPose (Jiang et al., 2023) | YOLOv5 | 23.9 | 11.6 | **31.9** | 17.0 | **32.3** | 17.9 |
| ViTPose (Xu et al., 2022) | YOLOv5 | **25.0** | **13.8** | 31.6 | **18.6** | 32.1 | **19.2** |
| DEKR$^\dagger$ + HRNet (Sun et al., 2019) | - | 19.2 | 11.8 | 23.5 | 17.1 | 24.3 | 17.9 |

Table 4: The results of 2D rose localization measured by the LKS-based index. AP@X° refers to the average precision and recall under the condition that the angular $\theta$ is less than X°. DEKR$^\dagger$ is an internal design inspired by recent decoupled keypoint regression strategies (Geng et al., 2021).

we make specific adjustments and improvements to the evaluation metric to better reflect the characteristics of our dataset, and conduct evaluations accordingly. We describe our proposed LKS-based evaluation index as follows:

The existing evaluation metric, defined by COCO (Lin et al., 2014), is inspired by the object detection task, which utilizes Object Keypoint Similarity (OKS) to measure the similarity between the ground truth and the predicted objects. We adhere to this definition and propose an extension by introducing a Line-based Keypoint Similarity (LKS) metric tailored for the Rosetum3D dataset, which is suitable for linear object structures.

For each object, the ground truth keypoints are denoted by:

$$[x_1, y_1, v_1, \ldots, x_k, y_k, v_k] \tag{2}$$

where $(x_i, y_i)$ denotes the keypoint locations, and $v_i$ is the visibility flag defined as:

$$v_i = \begin{cases} 0, & \text{not labeled} \\ 1, & \text{labeled but not visible} \\ 2, & \text{labeled and visible} \end{cases} \tag{3}$$

We also define the length $L$ as the Euclidean distance between the two farthest keypoints of the ground truth line segment.

Additionally, we introduce the angle $\theta$, defined as the angle between the rays of the predicted and ground truth objects. Both predicted and ground-truth rays are fitted by applying least-squares linear regression to all visible keypoints, constrained to pass through the first visible keypoint.

Let $d_i$ denote the Euclidean distance between each corresponding ground truth and detected keypoint, and $v_i$ the visibility flag of the ground truth (the detector's predicted $v_i$ are not used). Each $d_i$ is passed through an unnormalized Gaussian function with standard deviation $L\kappa_i$, where $\kappa_i$ is a per-keypoint constant controlling the falloff, which yields a keypoint similarity for each keypoint ranging between 0 and 1.

The final similarity is computed as the average of the similarities between the two keypoints considered. Predicted keypoints with $v_i = 0$ (not labeled) are skipped in the calculation, and the next labeled keypoint is used instead.

The **Line-based Keypoint Similarity (LKS)** is computed and included in matching only if the angle $\theta$ meets a preset threshold:

| Method | Encoder | Lower is better ↓ | | | Higher is better ↑ | | |
|--------|---------|--------|------|------------|-----------|-----------|-----------|
| | | AbsRel | RMSE | $\log_{10}$ | $\delta_1$ | $\delta_2$ | $\delta_3$ |
| **Zero-shot** | | | | | | | |
| IEBins (Shao et al., 2025) | Swin-Large | 0.554 | 1.655 | 0.211 | 0.241 | 0.476 | 0.716 |
| ZoeDepth (Bhat et al., 2023) | ViT-L | 0.591 | 2.186 | 0.270 | 0.266 | 0.468 | 0.618 |
| Depth Anything (Yang et al., 2024) | ViT-L | **0.498** | **1.092** | **0.163** | **0.373** | **0.664** | **0.831** |
| **With-Train** | | | | | | | |
| DPT (Ranftl et al., 2021) | VIT-B | 0.211 | 0.825 | 0.081 | 0.738 | 0.930 | 0.970 |
| DepthFormer (Li et al., 2022a) | Swin-Large | 0.181 | **0.688** | 0.067 | 0.797 | **0.950** | **0.978** |
| BinsFormer (Li et al., 2022b) | Swin-Large | **0.165** | 0.734 | **0.066** | **0.822** | 0.946 | 0.974 |

Table 5: The results of the monocular depth estimation task on the Rosetum3D dataset. For zero-shot evaluation, the models are trained on NYU Depth V2 (Silberman et al., 2012) and tested on the Rosetum3D dataset without fine-tuning.

$$\text{LKS} = \begin{cases} \dfrac{\exp\left(-\frac{d_1^2}{2L^2\kappa_1^2}\right)\cdot\delta(v_1>0) + \exp\left(-\frac{d_k^2}{2L^2\kappa_k^2}\right)\cdot\delta(v_k>0)}{\delta(v_1>0)+\delta(v_k>0)}, & \text{if } \theta < \theta_\varepsilon \\ 0, & \text{otherwise} \end{cases} \quad (4)$$

In our evaluation, we set $\theta_\varepsilon$ thresholds at 1°, 3°, and 10°. The computation of Average Precision (AP) and Average Recall (AR) follows the original evaluation protocol used in keypoint detection tasks. The results evaluated by the LKS-based index are presented in Table 4. Here, the top-down approaches (RTMPose, VitPose) outperform the bottom-up approaches (DEKR$^\dagger$ + HRNet), a result similar to those in human pose estimation, which demonstrates the effectiveness of the LKS-based evaluation index. We do not show the results of YOLOXPose (Maji et al., 2022) because that model performs poorly in predicting the first and last points of the rose object, resulting in few effective predictions.

## 4.3 MONOCULAR DEPTH ESTIMATION

Leveraging structured-light depth sensors during multi-scene data acquisition, we curate 33,019 precisely aligned RGB-D image pairs to establish a comprehensive dataset for monocular depth estimation. To benchmark agricultural depth perception challenges, we evaluate state-of-the-art monocular depth estimation methods using dual protocols: zero-shot inference on pretrained models and fine-tuning evaluation with our training split. Quantitative results across key metrics (e.g., AbsRel, RMSE, $\delta < 1.25$) are systematically compared in Table 5, while some qualitative comparison results are shown in the Appendix A.5. The results show that the fine-tuning procedure can significantly improve performance.

## 4.4 3D ROSE LOCALIZATION

In the 3D keypoint localization task, we reconstruct 3D keypoints by combining the inferred 2D keypoints with depth estimates from a depth prediction model. Evaluation is conducted on ground-truth 3D keypoint data, which is derived from manual annotations and corresponding ground-truth depth maps. A top-down keypoint detection model is employed with the DETR object detector, where all keypoint prediction results are evaluated on a per-instance basis. Depth inference is performed using both zero-shot and trained depth models. All evaluated scenes are different from the depth estimation training dataset. Besides using the predicted depth map, we also compare the results by directly using the ground truth depth map, which is obtained from the RGB-D sensors.

When evaluating the results, we employ the Mean Per Joint Position Error (MPJPE), commonly used in Human Pose Estimation (HPE) evaluations (Zhang et al., 2022; Zhao et al., 2023a; Zhu et al., 2023). For evaluation, we select the predicted instance with the highest confidence score and its corresponding ground-truth instance, whose results are presented in Table 6. And some qualitative comparison results are shown in the Appendix A.6.

| 2D Keypoint Inferencer | Detector | Depth Inferencer | Encoder | MPJPE $\downarrow$ |
|---|---|---|---|---|
| ViTPose (Xu et al., 2022) | DETR | Depth Anything | ViT-L | 656.9 |
| | | BinsFormer | Swin-Large | 647.2 |
| | | Ground Truth | - | 556.8 |
| RTMPose (Jiang et al., 2023) | DETR | Depth Anything | ViT-L | 650.6 |
| | | BinsFormer | Swin-Large | *644.5* |
| | | Ground Truth | - | 558.4 |
| DEKR$^\dagger$ + HRNet (Sun et al., 2019) | - | Depth Anything | ViT-L | 669.4 |
| | | BinsFormer | Swin-Large | 654.3 |
| | | Ground Truth | - | **522.8** |
| YOLOXPose (Maji et al., 2022) | - | Depth Anything | ViT-L | 653.2 |
| | | BinsFormer | Swin-Large | 645.1 |
| | | Ground Truth | - | 589.3 |

Table 6: 3D rose localization results obtained by integrating the 2D keypoint inference model with the depth estimation model. Measured in MPJPE (lower is better). DEKR$^\dagger$ is an internal design inspired by recent decoupled keypoint regression strategies (Geng et al., 2021). *Italic* indicates the best results without extra information. **Bold** indicates the best results by introducing extra information (the ground-truth depth map).

#### 4.4.1 LKS3D-BASED EVALUATION INDEX

Inspired by our proposed LKS-based metric, we design a novel LKS3D-based evaluation index for evaluating 3D rose keypoint localization. Specifically, for each matched pair of predicted and ground-truth keypoint sets, we fit a spatial line using the least squares method.

Let the ground-truth keypoints be $\{\mathbf{g}_i\}_{i=1}^K$, and the predicted keypoints be $\{\mathbf{p}_i\}_{i=1}^K$.

We fit spatial lines $\mathcal{L}_g$ and $\mathcal{L}_p$ to the ground-truth and predicted points, respectively, using the least square method.

The length of the ground-truth segment is defined as

$$L = \|\mathbf{g}_K - \mathbf{g}_1\|_2 \tag{5}$$

and the average endpoint error between prediction and ground-truth is

$$D = \frac{1}{2} \left( \|\mathbf{p}_1 - \mathbf{g}_1\|_2 + \|\mathbf{p}_K - \mathbf{g}_K\|_2 \right). \tag{6}$$

We compute the angle $\theta$ between the two fitted lines by

$$\theta = \arccos \left( \frac{\mathbf{v}_g \cdot \mathbf{v}_p}{\|\mathbf{v}_g\|_2 \|\mathbf{v}_p\|_2} \right), \tag{7}$$

where $\mathbf{v}_g$ and $\mathbf{v}_p$ are the direction vectors of $\mathcal{L}_g$ and $\mathcal{L}_p$, respectively.

In this way, a prediction is considered accurate if both the normalized average endpoint error and the angle meet the thresholds:

$$\frac{D}{KL} < \tau_d \quad \text{and} \quad \theta < \tau_\theta, \tag{8}$$

where $\tau_d$ and $\tau_\theta$ are predefined thresholds for the normalized distance and angle. $K$ is a scalar that serves as a measure and is set to 2. Finally, the accuracies of different models are shown in Table 7. Here, we do not show the results of DEKR$^\dagger$ + HRNet (Sun et al., 2019) because that model performs poorly in predicting the first and last points of the rose object, resulting in few effective predictions.

Comparing the results of 2D rose keypoint localization (Table 3 and 4) and 3D rose keypoint localization (Table 6 and 7), top-down keypoint localization methods show better stability than bottom-up approaches on the Rosetum3D dataset, both in 2D and 3D.

## 5 CONCLUSION

This paper introduces Rosetum3D, a pioneering large-scale RGB-D dataset designed to overcome the critical data scarcity in floriculture automation by capturing occlusion-heavy rose cultivation

| 2D Keypoint Inferencer | Depth Inferencer | ACC@1 & 15° ↑ | ACC@1 & 30° ↑ | ACC@1 & 45° ↑ |
|---|---|---|---|---|
| ViTPose (Xu et al., 2022) | BinsFormer | 6.0 | 23.1 | *64.5* |
| | Ground Truth | 67.0 | 88.3 | **96.1** |
| RTMPose (Jiang et al., 2023) | BinsFormer | *10.4* | *29.3* | 63.7 |
| | Ground Truth | 65.0 | 88.9 | 94.9 |
| YOLOXPose (Maji et al., 2022) | BinsFormer | 9.6 | 27.8 | 63.0 |
| | Ground Truth | **68.2** | **91.0** | 95.2 |

Table 7: 3D rose localization results measures by LKS3D-based index, where ACC@$\tau_d$ & $\tau_\theta$ indicates the accuracy when the thresholds of the normalized distance and angle are set to $\tau_d$ and $\tau_\theta$. *Italic* indicates the best results without extra information. **Bold** indicates the best results by introducing extra information (the ground-truth depth map).

scenes across diverse varieties and growth stages using structured-light depth sensors, which enables biologically relevant annotations through our hybrid 2D-to-3D framework. We first manually label 2D bounding boxes and several semantic keypoints per rose and subsequently leverage depth maps to reconstruct their 3D positions, achieving accurate measurements. The Rosetum3D dataset establishes the first agricultural benchmark for occlusion-invariant vision tasks, including 2D object detection, 2D/3D rose localization, local feature matching, and depth estimation. We believe that Rosetum3D can boost the intelligence and automation of agriculture. We will release Rosetum3D openly to catalyze innovation, bridging the computer vision algorithms and agriculture.

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

# A APPENDIX

## A.1 ANNOTATION VALIDATION EXPERIMENT

In this section, we detail the annotation validation experiment designed to quantify the geometric fidelity of our 3D reconstruction pipeline. To verify that the inter-keypoint distances of rose stems derived from 2D annotations and depth backprojection align with physical measurements, we captured supplementary RGB-D sequences during data acquisition. Each sequence included a calibration ruler positioned parallel to the primary stem axis, within 0.5–1.0 m from the camera, ensuring clear visibility of metric markings. Some examples of these RGB images are shown in Figure 3. Using our standard annotation protocol, we labeled two precise 10 cm interval ticks on each ruler. The 3D Euclidean distance between these points was computed via depth-based backprojection. Results demonstrate a mean absolute error (MAE) of 0.14 cm ($\sigma = 0.15$ cm) across 137 measurements, details are shown in Table 8. This result validates our annotation framework's capability to support the precision of Rosetum3D annotations.

## A.2 COMPARISON WITH OTHER 3D VISION DATASETS

We conduct a comparative analysis between Rosetum3D and existing public 3D vision datasets, as shown in Table 9.

## A.3 LOCAL FEATURE MATCHING

We represent each scene using its RGB images as the complete data source, processing scenes independently with COLMAP for 3D reconstruction following a methodology analogous to MegaDepth (Li & Snavely, 2018). The resulting reconstructions serve as ground truth for feature matching. Leveraging this pipeline, we establish **Rosetum3D-1800** – a benchmark comprising 1,800 image pairs curated for testing – enabling zero-shot evaluation of existing feature matching methods. Quantitative results are reported in Table 10. While comparable in scale to existing feature matching benchmarks, Rosetum3D-1800 presents distinct challenges due to high similarity among local features in agricultural environments.

We visualize qualitative local feature matching results for state-of-the-art models, including DKM (Edstedt et al., 2023), RoMa (Edstedt et al., 2024b), PMatch (Zhu & Liu, 2023), and De-DoDe (Edstedt et al., 2024a), on an identical rose cultivation image pair from Rosetum3D-1800 in Figure 4. All models undergo zero-shot evaluation, preserving raw outputs to maintain unbiased performance assessment.

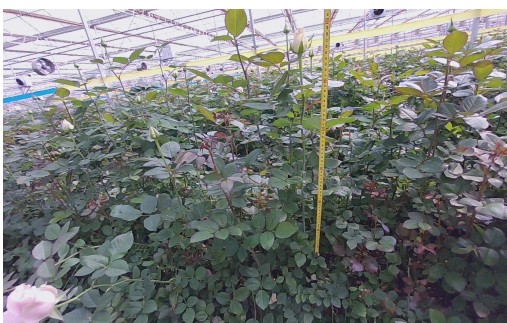

Figure 3: A sample of Annotation Validation with the ruler.

| Error(cm) | 0.00 - 0.15 | 0.15 - 0.30 | 0.30 - 0.45 |
|---|---|---|---|
| Num | 85 | 45 | 7 |

Table 8: Data Statistics of Annotation Validation.

| Dataset | Year | Scene Type | Size | Acquisition Methoh | Modality |
|---|---|---|---|---|---|
| NYU Depth V2 | 2012 | Indoor scenes | 0.5k scenes | RGB-D sensor | RGB-D |
| SUN RGB-D | 2015 | Indoor scenes | 10k images | RGB-D sensor | RGB-D |
| ScanNet | 2017 | Indoor scenes | 2,500k frames | RGB-D sensor | RGB-D |
| ScanNet++ | 2023 | Indoor scenes | 3,000k+ frames | RGB-D sensor | RGB-D |
| Human3.6M | 2014 | Human | 3,600k frames | Multi-view RGB + MoCap | RGB + 3D keypoints |
| KITTI | 2012 | Vehicle (driving) | 200k frames | Stereo RGB + LiDAR + GPS | RGB + LiDAR |
| nuScenes | 2019 | Vehicle (driving) | 1,400k frames | RGB + LiDAR + Radar | RGB + LiDAR + Radar |
| PandaSet | 2021 | Vehicle (driving) | 48k images & 16k LiDAR sweeps | RGB + LiDAR | RGB + LiDAR |
| Rosetum3D (ours) | 2025 | Rose scenes | 20k+ images & 20k+ depth maps | RGB-D sensor | RGB-D + 3D keypoints |

Table 9: Comparison between Rosetum3D and Other Public 3D Vision Datasets, including NYU Depth V2 (Silberman et al., 2012), SUN RGB-D (Song et al., 2015), ScanNet (Dai et al., 2017), ScanNet++ (Yeshwanth et al., 2023), Human3.6M (Ionescu et al., 2014), KITTI (Geiger et al., 2012), nuScenes (Caesar et al., 2020), and PandaSet (Xiao et al., 2021).

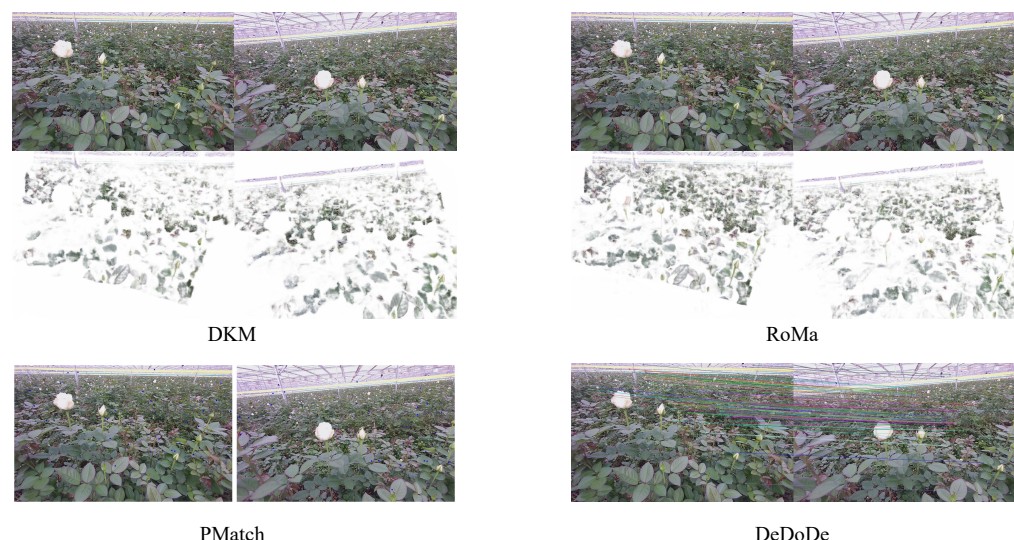

Figure 4: Comparison of Results Across Different Local Feature Matching Models on the Same Image Pair.

| Method | AUC@ → | 5° ↑ | 10° ↑ | 20° ↑ |
|---|---|---|---|---|
| DeDoDe (Edstedt et al., 2024a) | | 62.3 | 75.5 | 84.9 |
| DKM (Edstedt et al., 2023) | | 71.1 | 80.6 | 87.5 |
| PMatch (Zhu & Liu, 2023) | | 70.7 | 80.9 | **88.0** |
| RoMa (Edstedt et al., 2024b) | | **72.3** | **81.4** | 87.8 |

Table 10: The results of the local feature matching task on Rosetum3D-1800 with zero-shot evaluation. Measured in AUC (higher is better).

## A.4 VISUALIZATION OF 2D ROSE LOCALIZATION

In this section, we visualize fine-grained 2D ground-truth annotations alongside comparative results from leading keypoint localization models, as shown in Figure 5. The evaluation encompasses two top-down approaches (RTMPose Jiang et al. (2023) and ViTPose Xu et al. (2022)) and two bottom-up approaches (DEKR Geng et al. (2021) + HRNet Sun et al. (2019) and YOLOXPose Maji et al. (2022))

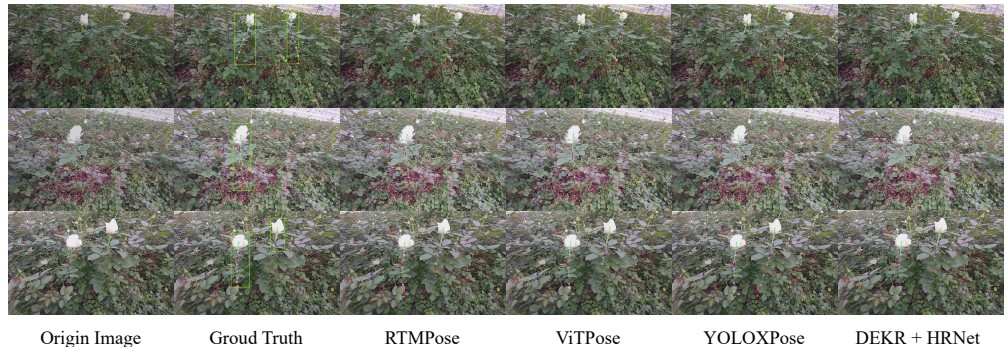

Origin Image     Groud Truth     RTMPose     ViTPose     YOLOXPose     DEKR + HRNet

Figure 5: Comparison of 2D Rose Localization Results.

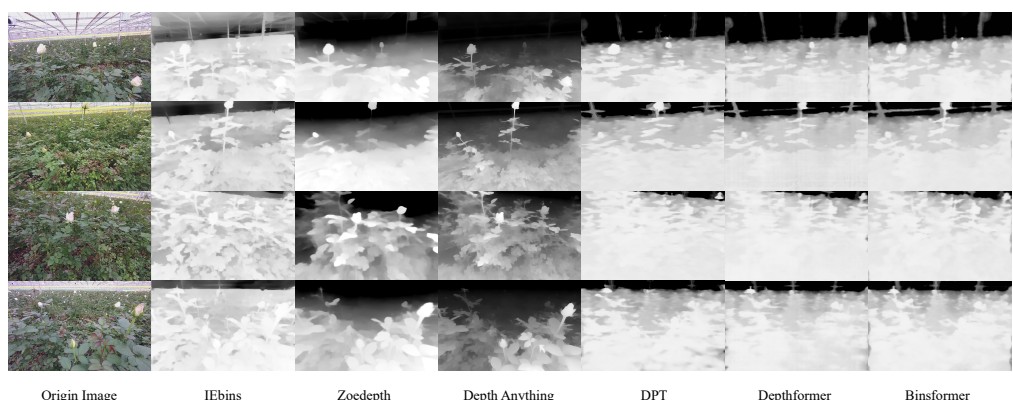

Origin Image     IEbins     Zoedepth     Depth Anything     DPT     Depthformer     Binsformer

Figure 6: Comparison of Depth Estimation Results.

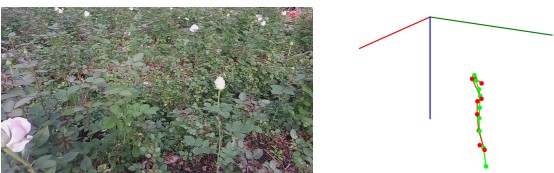

Figure 7: A 3D Rose Localization case. The left shows the original image, and the right shows the spatial reconstruction results from the keypoints. Green result indicates the ground truth, and red result represents the results inferred by VitPose (Xu et al., 2022) + Depth Anything (Yang et al., 2024).

## A.5 VISUALIZATION OF MONOCULAR DEPTH ESTIMATION

Figure 6 visualizes some monocular depth estimation results from representative models evaluated on Rosetum3D, including three Zero-shot methods (IEbins (Shao et al., 2025), Zoedepth (Bhat et al., 2023), and Depth Anything (Yang et al., 2024)) and three Fine-tuned approaches (DPT (Ranftl et al., 2021), Depthformer (Li et al., 2022a), and Binsformer (Li et al., 2022b)). Ground-truth depth maps from synchronized RGB-D sensors are provided as reference. All depth maps are rendered in grayscale (brightness indicates proximity).

## A.6 Visualization of 3D Rose Localization

We visualize ground-truth and predicted 3D keypoints within a unified camera coordinate system (origin at top-left corner) in Figure 7, with axes color-coded as:

- Red: Z-axis (depth, perpendicular to image plane).
- Green: X-axis (horizontal image direction).
- Blue: Y-axis (vertical image direction).

The XY-plane aligns with the camera's imaging plane, enabling direct correlation between pixel coordinates and spatial positions for precision stem localization in rose harvesting contexts.

## B Details of LLM Usage of This Paper

We use some LLMs in the following two ways:

- To aid and polish the writing;
- To retrieval and discovery (e.g., finding related work).

We use the following LLMs: DeepSeek (DeepSeek, 2025), ChatGPT (OpenAI, 2025), and Google Gemini (Google, 2025).

