# OpenReview forum: "Rosetum3D: A Large-Scale 3D Vision Dataset from Preharvest Roses"
_ICLR.cc/2026/Conference — Submitted to ICLR 2026_

### Official Review · Reviewer_KC5z · 2025-10-26

**Soundness:** 3
**Presentation:** 3
**Contribution:** 3
**Rating:** 4
**Confidence:** 3

**Summary:**

This paper introduces Rosetum3D, a dataset for 3D vision in rose cultivation captured using RGB-D cameras in commercial greenhouses. The dataset includes multi-view sequences with 2D bounding boxes and botanically-defined keypoints, which are backprojected to 3D using depth maps.

**Strengths:**

- Addresses an underserved domain (agricultural 3D vision) with clear practical applications for robotic harvesting and yield prediction
- Comprehensive multi-view RGB-D capture with synchronized depth, enabling multiple downstream tasks
- Thoughtful annotation protocol using 2D labeling with depth backprojection, avoiding fragile 3D reconstruction in occluded environments
- Rigorous validation methodology (ruler-based measurement with 0.14 cm MAE)

**Weaknesses:**

- The scientific contribution is limited to data collection. While valuable, the paper doesn't introduce novel methods, algorithms, or theoretical insights. For a top-tier venue like ICLR, datasets typically need to enable fundamentally new research directions or demonstrate surprising findings.

- The scale claim ("large-scale") is somewhat overstated. With 20k+ images, Rosetum3D is substantial but modest compared to datasets like ScanNet (2.5M views) or nuScenes (1.4M frames) mentioned in comparisons. The number of annotated keypoints or 3D structures isn't clearly stated.

- Limited analysis of what makes rose cultivation uniquely challenging. The paper mentions occlusions and morphological variation but doesn't quantify these challenges or demonstrate that existing methods fail specifically due to domain characteristics rather than lack of training data.

- The multi-task benchmarking is comprehensive but shallow. Baseline experiments mostly apply off-the-shelf methods without domain adaptation or analysis of failure modes. What vision problems are specific to roses versus other plants or cluttered scenes?

**Questions:**

1. Can you provide more detailed statistics: total number of annotated keypoints, distribution across growth stages and varieties, occlusion severity metrics?

2. What is the inter-annotator agreement for 2D keypoint annotations? Have you measured this systematically?

3. How does depth quality vary with distance from camera? Are there systematic errors in backprojection at different depths or viewing angles?

4. You mention "botanically defined" keypoints—can you provide botanical references or expert validation for these choices? How do they map to actual quality grading criteria used in industry?

5. Have you compared RGB-D backprojection with alternative 3D supervision strategies (e.g., multi-view triangulation, SfM)? Why is backprojection definitively better for this domain?

6. Will you release code for the annotation pipeline and evaluation protocols? What about pretrained models?

7. The ruler validation is helpful, but have you validated against other ground truth (e.g., manual measurements of actual stem lengths)?

---

> ### Author Response · Authors · 2025-11-21
> **Response to Reviewer KC5z (Part1)**
>
> We sincerely thank you for your insightful and professional feedback!
> ## The scientific contribution is limited to data collection. (Weakness #1)
> - The novelty of this paper mainly focuses on the novel dataset on pre-harvested roses, and we also transfer the methods of human keypoint localization into the rose objects.
> ## The scale claim ("large-scale") is somewhat overstated. And more detailed statistics. (Weakness #2 & Question #1)
> - The controlled environment of a constant-temperature greenhouse, with its regulated temperature and lighting systems, minimizes the impact of external seasonal variations, temperature fluctuations, and light conditions. The dataset we have compiled already encompasses a variety of rose varieties, with differences manifested in characteristics such as color, petal and sepal morphology, and the presence or absence of thorns on the stem. Each scene corresponds to a distinct rose variety, thereby inherently incorporating a degree of generalization. In comparison to existing datasets primarily focused on crop cultivation scenarios[R1-R6], Rosetum3D exhibits a relatively extensive scale.
> - A more detailed statistical summary of the data (as of the time of manuscript submission) has been added to Table 1 in the main text. Details are as follows: For the 2D keypoint localization task, the training and test sets consist of 17 and 3 scenes, 17,924 and 3,190 images, 20,759 and 6,089 object instances, 366,831 and 54,801 keypoints, respectively. Among these keypoints, we divide the keypoints in 3 categories by the visibility (Section 4.2.1): the numbers of visible keypoints are 281,799 and 42,728 ($v_i=2$), partially occluded but detectable keypoints ($v_i=1$) are 1,758 and 610, and invisible keypoints ($v_i=0$) are 83,294 and 11,463, respectively.
> - [R1] A. Conn et al., High-resolution laser scanning reveals plant architectures that reflect universal network design principles. Cell systems, 2017.
> - [R2] A. Conn et al., A statistical description of plant shoot architecture. Current Biology, 2017.
> - [R3] A. Conn et al., Network trade-offs and homeostasis in arabidopsis shoot architectures. PLOS Computational Biology, 2019.
> - [R4] D. Schunck et al. Pheno4d: A spatio-temporal dataset of maize and tomato plant point clouds for phenotyping and advanced plant analysis. Plos one, 16:e0256340, 2021.
> - [R5] Y. Sun et al., Soybean-mvs: Annotated three-dimensional model dataset of whole growth period soybeans for 3d plant organ segmentation. Agriculture, 2023.
> - [R6] B. Franchettiet al., Vision based modeling of plants phenotyping in vertical farming under artificial lighting. Sensors, 2019
> ## What vision problems are specific to roses versus other plants or cluttered scenes? (Weakness #4)
> - The target structure of roses presents unique characteristics, being distinctly elongated and exhibiting a notably high aspect ratio as objects for detection. This distinguishes them significantly from other crops and common detection targets. Consequently, transferring recognition algorithms developed for other targets (such as humans) to the rose domain requires a careful evaluation of their adaptability, which constitutes a primary focus of this work. We also hope that Rosetum3D will help advance research not only in rose-specific recognition but also in general object detection.
> ## What is the inter-annotator agreement for 2D keypoint annotations? Have you measured this systematically? (Question #2)
> - All annotators underwent multiple training sessions to standardize criteria, such as the determination of occlusion and visibility. Furthermore, supervisors conducted random sampling and review of each annotator's work. Annotations that failed to meet the required standards were revised to ensure compliance with predefined quality criteria.

---

> ### Author Response · Authors · 2025-11-21
> **Response to Reviewer KC5z (Part2)**
>
> ## How does depth quality vary with distance from camera? Are there systematic errors in backprojection at different depths or viewing angles? (Question #3)
> - In our data acquisition scenario, distances that are either too far or too close can indeed lead to a degradation in the depth quality captured by the sensors. We maintained an optimal distance as much as possible during data collection (see Section 3.1.3 of the main text) and incorporated a validation procedure (see Supplementary Material A.3) to calculate the data acquisition error at this distance. While systematic errors in the backprojection processing are inevitable, we believe these errors can be quantified and optimized.
> ## "Botanically defined" keypoints. (Question #4)
> - The overground structure of a rose flower can be botanically categorized into the corolla, sepals, pedicel, stem, and leaves, as documented in botanical references such as [R1] and [R2]. Among these, the leaves are not situated along the central axis and are therefore excluded from keypoint annotation. The remaining components exhibit distinct boundaries, which we adopt as the keypoints for annotation. On this basis, two additional keypoints, including the Corolla Center and the Median Point of the plant, are selected due to their high visibility. Furthermore, several auxiliary inter-points are incorporated to support the annotation framework, collectively establishing a comprehensive keypoint annotation system for rose plants.
> - [R1] Ray F. Evert and Susan E. Eichhorn, Raven Biology of Plants, Eighth Edition, 2008.
> - [R2] Ray F. Evert, Esau's Plant Anatomy: Meristems, Cells, and Tissues of the Plant Body: Their Structure, Function, and Development, 3rd Edition Ray F. Evert, 2006.
> ## Have you compared RGB-D backprojection with alternative 3D supervision strategies? (Question #5)
> - We previously attempted to use Structure-from-Motion (SfM) for 3D reconstruction, employing advanced tools such as COLMAP. However, the results were not satisfactory. This is primarily because the images contain complex foreground and background elements, along with a large number of similar features within a single image (e.g., green leaves appearing at different positions in the same image), which can be misidentified during feature matching and severely compromise camera pose estimation. Therefore, in this paper, we adopted a single-view RGB-D back-projection approach to directly obtain the reconstruction results. Our future work will include exploring multi-view reconstruction methods.
> ## Will you release code for the annotation pipeline, evaluation protocols, and pretrained models?(Question #6)
> - Indeed, we are continuously organizing and optimizing the codebase, and the corresponding pre-trained weights have been preserved.
> ## The ruler validation is helpful, but have you validated against other ground truth. (Question #7)
> - In principle, measuring the actual length of a rose stem has no difference from measuring a fixed segment on a ruler. Therefore, we consider the ruler validation to be already enough for verifying the accuracy of the back-projection and have thus refrained from conducting additional measurements on real rose stems.

---

> > ### Comment · Reviewer_KC5z · 2025-11-25
> >
> > Thank you for your response. I appreciate the clarifications, but my main concerns regarding limited methodological novelty and lack of deeper analysis remain, so I will maintain my initial score.

---

> > > ### Author Response · Authors · 2025-11-29
> > > **Response to Reviewer KC5z**
> > >
> > > Thank you for your review!
> > >
> > > We truly appreciate your valuable feedback and suggestions. We will continue to build the Rosetum3D dataset and improve our work in the future.
> > >
> > > Additionally, to give more details of our dataset, we have shared some data examples from Rosetum3D via the following anonymous link: https://anonymous.4open.science/r/Rosetum3D-E02C/. The samples include data from 5 different scenes, with 100 samples per scene. Each sample consists of an RGB image, a depth map, and annotations for rose bounding boxes and keypoints.

---

### Official Review · Reviewer_bjqL · 2025-10-31

**Soundness:** 3
**Presentation:** 3
**Contribution:** 2
**Rating:** 4
**Confidence:** 3

**Summary:**

This paper introduces Rosetum3D, a new large-scale RGB-D dataset collected in commercial rose greenhouses for 3D computer vision applications in floriculture. The data is captured across diverse rose varieties and growth stages, aiming to capture the structural variability and heavy occlusions typical of real cultivation environments. It contains synchronised RGB-D sequences that are carefully annotated with 2D bounding boxes and botanically defined 3D keypoints, which contain geometric and morphological details of rose plants.

The authors validate their annotation accuracy through quantitative measurements and propose two new evaluation metrics, LKS and LKS3D, designed for the linear structure of rose stems. They benchmark several standard tasks, including object detection, keypoint localisation, depth estimation, and local feature matching, using both CNN and transformer-based models. Overall, Rosetum3D is intended to serve as a foundational resource for research on rose phenotyping, growth monitoring, and robotic harvesting.

**Strengths:**

This paper tackles a clear and underexplored problem in floriculture vision, focusing on pre-harvest rose monitoring where dense occlusions and fine structural details make visual perception particularly challenging.

1. The dataset collection and annotation process are carefully designed, featuring botanically defined 3D keypoints and a quantitative validation study that demonstrates strong geometric accuracy (0.14 cm MAE).

2. The dataset is well constructed and diverse, covering multiple rose varieties and growth stages, and it supports several relevant vision tasks, including detection, keypoint localisation, and depth estimation.

3. The authors also make improvements to the OKS-based evaluation metric, introduce LKS-based evaluation metric, and establish baseline benchmarks using both CNN and transformer-based architectures.

**Weaknesses:**

1. While the newly proposed dataset and evaluation design are valuable, the paper does not introduce a new learning method or theoretical result. As a result, the novelty lies more in the resources than in the methodology, which may fall short of ICLR’s expectations for algorithmic innovation.

2. Scope is narrower than comparable plant datasets; for example, the Crops3D (Zhu et al., 2024) dataset spans multiple species and field conditions, while PlantGaussian (Shen et al., 2025) models plants across time and scenes. In contrast, Rosetum3D’s rose-only focus limits generalisation. A small cross-species transfer experiment or adding a subset from another flower or crop would strengthen the case for broader impact.

3. The benchmarks provide broad task coverage detection, 2D/3D keypoints, depth estimation, and feature matching, but the analysis stays mostly at a summary level. It’s not very clear, why models succeed or fail?. For instance, while 2Dkeypoint AP and LKS/LKS3D results are reported, there’s no breakdown showing which keypoints perform poorly (e.g., corolla centre vs. stem base) or how performance varies with occlusion, scale, or viewpoint, even though the dataset distinguishes between visible and invisible keypoints. Similarly, there’s no quantitative discussion of scene or occlusion characteristics, and no cross-domain or transfer experiments to assess generalisation. The LKS and LKS3D metrics appear reasonable for stem structures, but without validation against human evaluation or downstream task relevance, their practical significance remains uncertain.

4. Although Rosetum3D contains around 21K RGB-D images, these were collected from only 20 greenhouse scenes (17 for training and 3 for testing). The overall image count is reasonable for a specialised domain, but the small number of distinct capture environments limits the dataset’s spatial and contextual diversity.

5. The zero-shot depth models pretrained on NYU show a clear drop in performance on Rosetum3D, which is expected, but the paper does not investigate the underlying reasons. Factors such as sensor differences from the structured-light RGB-D setup, the optical properties of rose petals and leaves, the mixed greenhouse lighting, and the short-range, occlusion-heavy geometry could all contribute to this gap. However, the paper provides no breakdown of errors by range, occlusion level, viewpoint, or material type, nor does it include any ablation studies or adaptation experiments. As a result, we see that performance decreases, but the work offers little understanding of why this happens or how it might be addressed.

**Questions:**

1. Analysis of failure cases: The benchmarks span several tasks, but the analysis remains limited. Could you provide more insight into where the models fail, for example, which keypoints are most affected by occlusion, lighting variations, or viewpoint changes?

2. Validation of new metrics: The new LKS and LKS3D metrics seem reasonable, but have you validated that they correspond to perceived or practical improvements, for example, through expert evaluation or correlation with 3D reconstruction accuracy?

3. Depth generalisation: The zero-shot depth models pretrained on NYU perform noticeably worse on Rosetum3D. Can you analyse the main factors contributing to this gap --  such as sensor characteristics, material properties, or short-range geometry -- and discuss potential adaptation strategies?

4. Cross-scene and cross-species generalisation: Have you tested how models trained on Rosetum3D generalise across different greenhouse scenes or to other flower or crop datasets? Such an experiment could clarify whether the dataset encourages transferable representations beyond roses.

Overall, the paper presents a well-executed, clearly documented dataset that addresses a meaningful gap in floriculture research. However, it suffers from a limited methodological innovation and relatively shallow analysis.

---

> ### Author Response · Authors · 2025-11-21
> **Response to Reviewer bjqL**
>
> We sincerely thank you for your insightful and professional feedback!
> ## Do not introduce a new learning method or theoretical resul. (Weakness #1)
> - The novelty of this paper mainly focuses on the novel dataset on pre-harvested roses, and we also transfer the methods of human keypoint localization into the rose objects.
> ## Scope is narrower than comparable plant datasets. (Weakness #2 & Question #4)
> - We have added an additional part "Plant Scene" in Section 2.1, which included several related works.
> - Currently, we have not tested models trained on the Rosetum3D dataset on other flower species. Even within roses alone, there exist dozens of varieties (differing in color, petal and sepal shapes, presence or absence of thorns on stems, etc.). During our data collection, the rose varieties differed across various scenes. Consequently, the design of the Rosetum3D dataset inherently incorporates a degree of generalization capability, as the training and test sets contain different varieties. Therefore, our research currently focuses on generalization across different rose varieties and has not yet considered extensibility to other species.
> ## Analysis of failure cases. (Weakness #3 & Question #1)
> - In practical scenarios, the parts of a rose below the flower are susceptible to occlusion, with the root region experiencing the most severe occlusion. Most models exhibit suboptimal performance in predicting this region. In our subsequent work, we will consider supplementing root region data as we expand the dataset scale. For keypoint annotation references, please see Figure 2 or Section 3.2.
> ## The small number of distinct capture environments limits the dataset's spatial and contextual diversity. (Weakness #4)
> - The controlled environment of a constant-temperature greenhouse, with its regulated temperature and lighting systems, minimizes the impact of external seasonal variations, temperature fluctuations, and light conditions. The dataset we have compiled already encompasses a variety of rose varieties, with differences manifested in characteristics such as color, petal and sepal morphology, and the presence or absence of thorns on the stem. Each scene corresponds to a distinct rose variety, thereby inherently incorporating a degree of generalization.
> ## Depth generalisation. (Weakness #5 & Question #3)
> - We have given extensive consideration to the depth estimation task and have also conducted research on the acquisition sensors (the specific devices used are detailed in Section 3.1.1). During our data collection process in rose cultivation environments, the overall performance of existing RGB-D cameras in capturing depth data proved inferior compared to common indoor scenarios. This is primarily attributable to the slender structure of floral stems, which often results in substantial depth disparities between the foreground and background. Furthermore, although the acquisition environment is indoors, the lighting conditions are more intense than those in typical indoor settings, and the large spatial extent of the scene may exceed the effective range of the sensors at farther distances.
> - In contrast, the NYU dataset mainly consists of indoor scenes where both foreground and background fall within the sensor's operational range with minimal depth variation, and object boundaries are well-defined. Overall, zero-shot depth estimation models pre-trained on the NYU dataset are unlikely to perform satisfactorily on Rosetum3D. Subsequent work will focus on exploring more effective transfer learning strategies to better adapt depth estimation models trained on existing datasets to the Rosetum3D environment.
> ## Validation of new metrics. (Question #2)
> - Currently, the correlation between our proposed LKS/LKS3D metrics and 3D reconstruction performance has not yet been validated. The introduction of Rosetum3D is designed to address the lack of visual datasets and benchmarks in the agricultural flower domain. Essentially, our objective is to evaluate the similarity between the line segments fitted from predicted keypoints and the ground-truth (GT) segments. The LKS/LKS3D metrics align with our intended design to assess the prediction accuracy of linear structures.

---

> ### Comment · Reviewer_bjqL · 2025-11-25
> **Ansawer to Authors**
>
> Dear Authors,
>
> Thanks a lot for your detailed answer. I understand both the difficulties and the relevance of a "novel dataset on pre-harvested roses". However, to defend the dataset tout court, it would have been nice to upload some examples of your dataset via an anonymous link, which would have required giving details on release (see Reviewer e3e8). In the absence of examples, it is natural for reviewers to insist on methodology and comparisons.

---

> > ### Author Response · Authors · 2025-11-29
> > **Response to Reviewer bjqL**
> >
> > Thank you for your advice!
> >
> > To give more details of our dataset, we have shared some data examples from Rosetum3D via the following anonymous link: https://anonymous.4open.science/r/Rosetum3D-E02C/. The samples include data from 5 different scenes, with 100 samples per scene. Each sample consists of an RGB image, a depth map, and annotations for rose bounding boxes and keypoints.

---

### Official Review · Reviewer_eRe8 · 2025-11-01

**Soundness:** 2
**Presentation:** 2
**Contribution:** 2
**Rating:** 4
**Confidence:** 3

**Summary:**

This paper introduces Rosetum3D, a multi-view RGB-D dataset captured in operational greenhouses, enabling 2D/3D localization and phenotyping of pre-harvest roses. The authors annotate each rose with a bounding box and nine botanically meaningful keypoints (five primary + four interpolated), then back-project these to 3D via synchronized depth, avoiding brittle full 3D reconstruction in dense, occlusion-heavy canopies. The capture setup uses Orbbec structured-light stereo sensors (1280×800 @ 30 Hz) with a prescribed walking protocol and calibration pipeline; annotation fidelity is validated with ruler measurements showing 0.14 cm MAE across 137 cases. The dataset supports benchmarks for detection, 2D/3D keypoint localization, monocular depth estimation, and local feature matching (a Rosetum3D-1800 split), and proposes new line-aware metrics (LKS/LKS3D) better aligned with stem-like morphology. Baselines indicate moderate detection (e.g., YOLOv5 AP≈35) and that top-down keypoint methods excel when perfect boxes are provided; fine-tuned depth models substantially outperform zero-shot ones on this domain. If released, Rosetum3D would fill a notable gap in floriculture vision and provide a concrete testbed for pre-harvest grading and robotic harvesting research.

**Strengths:**

- Well-motivated, carefully engineered data pipeline for a genuinely under-served application domain (pre-harvest floriculture), with practical choices (structured-light RGB-D, greenhouse-robust protocol) and thorough calibration details.
- Annotation and metric design grounded in plant morphology, including five biologically defined keypoints plus inter-points, and the LKS/LKS3D metrics that emphasize line orientation/endpoints, appropriate for stem-like structures.
- Comprehensive benchmarks spanning detection, 2D/3D keypoints, depth, and feature matching (Rosetum3D-1800), with both zero-shot and fine-tuned evaluations and clear takeaways (detector-limited top-down pipelines; fine-tuned depth wins).

**Weaknesses:**

- Inconsistent dataset statistics: the text states 66,848 annotated rose objects, while Table 1 totals 46,848—a substantial discrepancy that needs reconciliation and suggests QA issues in reporting.
- Scale and diversity may be limited relative to the “large-scale” claim, as only 20 scenes (~21k images ) from two camera models, one capture protocol (single walking trajectory and tilt), and one crop in one environment type are used, which may constrain generalization. Clarify sites, varieties, growers, and capture days/seasons.
- Metric clarity and robustness: LKS primarily uses the first/last keypoints, along with an angle threshold. However, results for some models hinge on endpoint prediction, raising concerns about metric sensitivity and potential gaming. LKS3D introduces normalized endpoint error with an unexplained constant (K=2) and threshold choices (e.g., “ACC@1&15°”), but units/interpretation are underspecified. Provide ablations and rationale.
- Missing or underspecified release details: no license, access policy, privacy/compliance statement with the commercial greenhouses, or a validation split (train/test only), which hinders reproducibility and fair model selection.

**Questions:**

- Which figure is correct for the total number of annotated objects: 46,848 (Table 1) or 66,848 (main text)? Please provide a cleaned statistics table (scenes, images, objects, keypoints, and invisibility rates) for training, validation, and testing.
- For LKS/LKS3D, could the authors justify the endpoint focus, angle thresholds, and the constant (K)? Have the authors evaluated robustness to endpoint mis-annotation or to varying which keypoints define the “line”?

---

> ### Author Response · Authors · 2025-11-21
> **Response to Reviewer eRe8**
>
> We sincerely thank you for your insightful and professional feedback!
> ## Typo about the total number. (Weakness #1 & Question #1)
> - Thank you for pointing this out. We will correct the erroneous figure "66,848" in the main text to "46,848" as presented in Table 1, and the corresponding revision has been made. We are continuously collecting and annotating data to expand the scale of our dataset. A more detailed statistical summary of the data (as of the time of manuscript submission) has been added to Table 1 in the main text.
> - Details are as follows: For the 2D keypoint localization task, the training and test sets consist of 17 and 3 scenes, 17,924 and 3,190 images, 20,759 and 6,089 object instances, 366,831 and 54,801 keypoints, respectively. Among these keypoints, we divide the keypoints in 3 categories by the visibility (Section 4.2.1): the numbers of visible keypoints are 281,799 and 42,728 ($v_i=2$), partially occluded but detectable keypoints ($v_i=1$) are 1,758 and 610, and invisible keypoints ($v_i=0$) are 83,294 and 11,463, respectively. Currently, no validation set has been defined; all models are trained on the training set and evaluated on the test set. A validation set will be introduced in subsequent expansions as the dataset grows.
> ## Scale and diversity may be limited relative to the "large-scale" claim. (Weakness #2)
> - The controlled environment of a constant-temperature greenhouse, with its regulated temperature and lighting systems, minimizes the impact of external seasonal variations, temperature fluctuations, and light conditions. The dataset we have compiled already encompasses a variety of rose varieties, with differences manifested in characteristics such as color, petal and sepal morphology, and the presence or absence of thorns on the stem. Each scene corresponds to a distinct rose variety, thereby inherently incorporating a degree of generalization. In comparison to existing datasets primarily focused on crop cultivation scenarios[R1-R6], Rosetum3D exhibits a relatively extensive scale.
> - [R1] A. Conn et al., High-resolution laser scanning reveals plant architectures that reflect universal network design principles. Cell systems, 2017.
> - [R2] A. Conn et al., A statistical description of plant shoot architecture. Current Biology, 2017.
> - [R3] A. Conn et al., Network trade-offs and homeostasis in arabidopsis shoot architectures. PLOS Computational Biology, 2019.
> - [R4] D. Schunck et al. Pheno4d: A spatio-temporal dataset of maize and tomato plant point clouds for phenotyping and advanced plant analysis. Plos one, 16:e0256340, 2021.
> - [R5] Y. Sun et al., Soybean-mvs: Annotated three-dimensional model dataset of whole growth period soybeans for 3d plant organ segmentation. Agriculture, 2023.
> - [R6] B. Franchettiet al., Vision based modeling of plants phenotyping in vertical farming under artificial lighting. Sensors, 2019
>
> ## LKS/LKS3D robustness evaluation. (Weakness #3 & Question #2)
> - Regarding the LKS/LKS3D evaluation metrics, due to the structural characteristics of roses, our intention is to propose a linear metric to assess the performance of keypoint localization algorithms on Rosetum3D. Essentially, we aim to evaluate the similarity between the line segments fitted from the predicted keypoints and the ground-truth (GT) segments. Therefore, we select the visible endpoints and the included angle to assess them. All parameters are chosen as appropriate values after testing to achieve the desired effect and to differentiate the performance of various models.
> - For the 2D keypoint task, our metric effectively represents model performance. However, for the 3D keypoint task, the accuracy of the 3D keypoints largely depends on the precision of the depth information. In our scenario, errors in 2D keypoint prediction can lead to significant depth inaccuracies, resulting in relatively poor stability (such as depth loss or outliers for endpoint predictions). We will conduct further tests and refinements on the LKS/LKS3D evaluation metrics to address these issues.
> ## Missing or underspecified release details. (Weakness #4)
> - In accordance with the anonymity requirements for submission, documents such as the license, access policy, and privacy/compliance statements have been temporarily omitted. These materials will be made available upon acceptance of the manuscript.
> - Currently, no validation set has been defined; all models are trained on the training set and evaluated on the test set. A validation set will be introduced in subsequent expansions as the dataset grows.

---

### Official Review · Reviewer_6Ccy · 2025-11-03

**Soundness:** 3
**Presentation:** 3
**Contribution:** 2
**Rating:** 4
**Confidence:** 4

**Summary:**

The paper introduces Rosetum3D, a large-scale RGB-D dataset for 3D vision in rose cultivation, targeting pre-harvest quality grading and growth monitoring in greenhouses. Data are collected with structured-light RGB-D cameras (Orbbec Gemini 335L/336L) under realistic greenhouse conditions, producing 21k synchronized RGB-D frames with nearly 47k annotated rose instances. Each instance includes 2D bounding boxes and 9 biologically defined keypoints, which are back-projected into 3D using the corresponding depth maps. The paper also proposes two new evaluation metrics, LKS (Line-based Keypoint Similarity) and LKS3D, designed for linear plant structures. Rosetum3D serves as a benchmark for multiple tasks, including object detection, keypoint localization (2D/3D), monocular/multi-view depth estimation and feature matching. Evaluation is performed using standard models such as YOLOv5, ViTPose and BinsFormer. Annotation accuracy is validated using calibrated rulers inserted in the scene, showing a mean absolute 3D error of 0.14 cm.

**Strengths:**

1. The paper fills a gap in agricultural 3D vision by introducing a novel floriculture-oriented RGB-D dataset with dense 2D/3D annotations.
2. Data collection, calibration and annotation processes are described in detail and validated quantitatively.
3. The paper introduces some metrics inspired by human-pose estimation to assess the quality of 2D/3D keypoint localization considering the linear plant geometry. These metrics may also be considered in other keypoint localization tasks concerning plants.
4. The dataset is contributing to applications concerning robotic harvesting, phenotyping, and precision agriculture, bridging vision research with real-world greenhouse applications.

**Weaknesses:**

1. The methodological novelty introduced by this work is limited. The main contribution lies in the dataset itself and in the introduction of the pose-inspired metrics. It would also be important to introduce a baseline method as a reference for the tasks targeted by the dataset introduced.
2. The dataset focuses solely on roses. It would be interesting to discuss whether the acquisition method, the LKS/LKS3D and the conclusions generalize to other floriculture or crop species.
3. The LKS/LKS3D metrics introduced in this work should be further analyzed. For instance, a more detailed discussion on the justification of the angle-based metrics would be valuable, as well as a discussion on the comparison with the OKS/MPJPE metrics on which they are based. The latter would help to assess how the new metrics contribute to the assessment of the localization quality.
4. Although robotic harvesting and yield prediction are motivating applications, the paper does not show any relevant use case.
5. Some complementary related work as [R1], [R2], and [R3] can be considered to provide better context.

### Minor comments
- Section 3.3 title contains a typo: “Staistics”

[R1] Mertoğlu et al., “PLANesT-3D: A New Annotated Data Set of 3D Color Point Clouds of Plants”. In Signal Processing and Communications Applications Conference (SIU) (pp. 1-4), 2023

[R2] Dutagaci et al., “ROSE-X: an annotated data set for evaluation of 3D plant organ segmentation methods”. Plant methods, 16(1), 28, 2020

[R3] Franchetti et al., “Vision based modeling of plants phenotyping in vertical farming under artificial lighting”. Sensors, 19(20), 4378. 2019

**Questions:**

1. Have the authors evaluated the performance of models trained on ScanNet or Crops3D on Rosetum3D?
2. How does LKS/LKS3D correlate with standard metrics like OKS or MPJPE? Are the rankings consistent?
3. Is there any temporal or multi-view consistency annotation to enable sequence-based learning?

---

> ### Author Response · Authors · 2025-11-21
> **Response to Reviewer 6Ccy**
>
> We sincerely thank you for your insightful and professional feedback!
> ## For the typo.
> - We have corrected the typo in Section 3.3 title, thank you for pointing out the typo.
> ## The methodological novelty introduced by this work is limited. (Weakness #1)
> - The novelty of this paper mainly focuses on the novel dataset on pre-harvested roses, and we also transfer the methods of human keypoint localization into the rose objects.
> ## The dataset focuses solely on roses. (Weakness #2)
> - Currently, we have not tested models trained on the Rosetum3D dataset on other flower species. Even within roses alone, there exist dozens of varieties (differing in color, petal and sepal shapes, presence or absence of thorns on stems, etc.). During our data collection, the rose varieties differed across various scenes. Therefore, our research currently focuses on generalization across different rose varieties and has not yet considered extensibility to other species.
> ## Have the authors evaluated the performance of models trained on ScanNet or Crops3D on Rosetum3D? (Question #1)
> - We currently do not directly transfer or evaluate models trained on ScanNet or Crops3D to Rosetum3D. The primary reason is the significant differences between Rosetum3D and the aforementioned datasets in terms of appearance, structure, semantic categories, and task definitions. ScanNet consists of indoor household scenes, which differ substantially from our Rosetum3D dataset. Although Crops3D, like Rosetum3D, focuses on individual plants in the agricultural domain, the data in Crops3D are primarily collected from orderly arranged potted plants and crops, which differs from the environment captured in Rosetum3D. Rosetum3D is a dataset representing realistic rose growth conditions. Models trained on Rosetum3D are intended to predict rose positions and stem lengths before harvesting, thereby enabling pre-harvest quality estimation and supporting automated harvesting tasks in future, while these capabilities cannot be afforded by the Crops3D dataset, which is based on systematically arranged crops.
> ## How does LKS/LKS3D correlate with standard metrics like OKS or MPJPE? (Weakness #3 & Question #2)
> - In human pose and keypoint estimation tasks, the focus is typically on the positional prediction accuracy of all keypoints. However, in Rosetum3D dataset, due to the distinct structure of individual roses, we place greater emphasis on how well the predicted keypoints can be fitted into straight lines or segments. This approach better aligns with the biological characteristics of rose plants. Therefore, we propose LKS/LKS3D, shifting the evaluation emphasis towards the accuracy of the overall line or segment rather than the precision of each individual point. As a result, the ranking outcomes based on OKS and MPJPE will differ from those derived using LKS/LKS3D. Furthermore, linear evaluation and potential subsequent linear features are expected to be a key focus when applying Rosetum3D to downstream tasks.
> ## Is there any temporal or multi-view consistency annotation to enable sequence-based learning? (Question #3)
> - Our raw data acquisition process inherently captures temporal sequences. However, during the annotation phase, frames without detectable individuals are removed, and we also shuffle and recombine the remaining data. We will release the original image sequences (with temporal information). Our current work involves annotating correspondences of the same individual across different images for multi-view consistency, which is intended to support the stereo localization and re-identification researches.
> ## The paper does not show any relevant use case robotic harvesting and yield prediction. (Weakness #4)
> - The integration of robotic harvesting and yield prediction, while highly relevant to agricultural applications, falls outside the primary scope of this paper. Our central objective is to introduce and characterize a novel dataset designed for benchmarking recognition algorithms. Therefore, we have intentionally focused our presentation on the dataset's properties and its benchmarking capabilities, leaving the development of control algorithms for robotic harvesting or specific yield prediction models for future downstream research.
> ## Some complementary related work. (Weakness #5)
> - Thank you for providing these related works, we have added an additional part "Plant Scene" in Section 2.1, which included several related works.

---

### Author Response · Authors · 2025-11-21
**General Response to All Reviewers**

Thanks to all the reviewers for their constructive feedback. We have addressed all the concerns, responded to each point, and made revisions to the paper. The modified sections are marked in blue. We sincerely hope you recheck and reconsider your decision.

To give more details of our dataset, we have shared some data examples from Rosetum3D via the following anonymous link: https://anonymous.4open.science/r/Rosetum3D-E02C/. The samples include data from 5 different scenes, with 100 samples per scene. Each sample consists of an RGB image, a depth map, and annotations for rose bounding boxes and keypoints.

---

### Meta-Review · Area_Chair_kYPc · 2025-12-17

**Summary:**

# Decision

The paper introduces Rosetum3D, an RGB-D dataset for rose cultivation in greenhouse conditions. Each rose instance is annotated with a 2D bounding-box, as well as 9 biologically-defined keypoints (similar to joint keypoints defined in 2D/3D human pose estimation). The annotation accuracy is quantitatively evaluated. The authors further propose two new metrics corresponding to their 2D/3D rose-stem keypoints, and finally benchmark CNN- and Transformer-based algorithms on several standard tasks (plant detection, keypoint localization, depth estimation, etc.).

The reviewers acknowledge the thorough and well-detailed protocol presented by the authors, and the value of tackling the lack of data for precision/robotic agriculture. The reviewers also appreciate the idea of adapting keypoint annotations from human pose estimation to plant-morphology evaluation. However, they also point out the limited scope of the dataset (focus on roses only, in greenhouses, with ~20K frames) and unclear motivation. While comprehensive, the benchmarking also remains superficial (off-the-shelf methods with no domain adaptation, lack of in-depth discussion of results).

Therefore, while the paper clearly has merit, the decision is not to recommend acceptance. The authors are encouraged to consider the reviewers' comments when revising the paper for submission elsewhere.

------------
# Consolidated Reviews

## Strengths

### Valuable benchmark for under-represented application
- Relatively-large dataset with rich annotations [`6Ccy`].
- Applicability to under-served topics, e.g., precision/robotic agriculture [`6Ccy`, `eRe8`, `bjqL`, `KC5z`].
- Data diversity (rose varieties, growth stages) [`bjqL`].
- Introduction of new keypoint annotations, grounded in plant morphology [`eRe8`].
- Introduction of new domain-relevant metrics based on novel keypoint annotations [`6Ccy`, `eRe8`, `bjqL`].

### Thorough protocol
- Collection and annotation protocols well detailed [`6Ccy`, `eRe8`, `bjqL`, `KC5z`].
- Thorough validation [`bjqL`, `KC5z`].
- Quantitative benchmarking on the proposed tasks and data [`6Ccy`, `eRe8`, `bjqL`].



## Weaknesses

### Limited methodological contributions
- Lack of algorithmic novelty, with focus on dataset contribution [`6Ccy`, `bjqL`, `KC5z`].
- Lack of justification w.r.t. the definition of proposed metrics [`6Ccy`, `eRe8`].
- Lack of discussion of benchmarking results [`bjqL`, `KC5z`].
- Comprehensive but shallow benchmarking [`KC5z`].

### Narrow dataset
- Narrow-scope dataset,  solely focused on roses / greenhouse / etc. [`6Ccy`, `bjqL`].
- Over-stated "large-scale" claim, c.f. 20 scenes / 20K images [`eRe8`, `bjqL`, `KC5z`].
- Concrete use-cases not detailed in the paper [`6Ccy`, `KC5z`].

### Misc.
- Missing release details [`eRe8`].
- Few typos [`6Ccy`, `eRe8`]

**Reviewer Concerns:**

See above for summary of main concerns shared by reviewers.

Overall, the authors have provided extensive justifications for some of their scope/design choices (e.g., variability of rose appearances, definition of proposed metrics). They have also highlighted their focus on dataset contribution rather than methodological novelty, which is fair for this track.

However, their justifications for the size / variability of the dataset might be deemed insufficient, as well as their replies w.r.t. the superficiality of their benchmarking. E.g., the authors reference a few agricultural datasets [R1-R6] as examples of prior work with similar data scope/methodology, but those were published in agriculture/biology-focused venues, not ML-focused ones where one should expect more scrutiny over data statistics and algorithmic evaluation.

**Reviewer Scores:**

### Reviewer `6Ccy`
- **Original score:** 4
- **Score change:** likely to keep their score. The authors covered the concerns/questions, e.g., about the metrics ; but gave unsatisfying answers w.r.t. paper motivation and evaluation of models trained on other agricultural datasets.

### Reviewer `eRe8`
- **Original score:** 4
- **Score change:** might have increased their score to ~6, but likely to have been unresponsive (c.f. shallow review). The authors covered the reviewer's mostly superficial comments, but the reviewer might have been influenced by the other deeper reviews.

### Reviewer `bjqL`
- **Original score:** 4
- **Score change:** likely to keep their score. After the first replies, the reviewer still appeared to question the contributions.


### Reviewer `KC5z`
- **Original score:** 4
- **Score change:** kept their score, c.f. their own reply.

---

### Decision · Program_Chairs · 2026-01-26

Reject